# First Investigation of the Optimal Timing of Vaccination of Nile Tilapia (*Oreochromis niloticus*) Larvae against *Streptococcus agalactiae*

**DOI:** 10.3390/vaccines11121753

**Published:** 2023-11-24

**Authors:** Benchawan Kumwan, Anurak Bunnoy, Satid Chatchaiphan, Pattanapon Kayansamruaj, Ha Thanh Dong, Saengchan Senapin, Prapansak Srisapoome

**Affiliations:** 1Laboratory of Aquatic Animal Health Management, Department of Aquaculture, Faculty of Fisheries, Kasetsart University, 50 Paholayothin Road, Ladyao, Chatuchak, Bangkok 10900, Thailand; backbony17@gmail.com (B.K.); ffisarb@ku.ac.th (A.B.); pattanapon.k@ku.th (P.K.); 2Center of Excellence in Aquatic Animal Health Management, Department of Aquaculture, Faculty of Fisheries, Kasetsart University, 50 Paholayothin Road, Ladyao, Chatuchak, Bangkok 10900, Thailand; 3Department of Aquaculture, Faculty of Fisheries, Kasetsart University, Bangkok 10900, Thailand; satid.ch@ku.th; 4Aquaculture and Aquatic Resources Management, Department of Food Agriculture and Bioresources, School of Environment, Resources and Development, Asian Institute of Technology, Pathum Thani 12120, Thailand; htdong@ait.ac.th; 5National Center for Genetic Engineering and Biotechnology (BIOTEC), National Science and Technology Development Agency (NSTDA), Pathum Thani 12120, Thailand; 6Fish Health Platform, Center of Excellence for Shrimp Molecular Biology and Biotechnology (Centex Shrimp), Faculty of Science, Mahidol University, Bangkok 10400, Thailand

**Keywords:** Nile tilapia, *Oreochromis niloticus*, *Streptococcus agalactiae*, formalin-killed vaccine, immunocompetence

## Abstract

To investigate early immune responses and explore the optimal vaccination periods, Nile tilapia at 1, 7, 14, 21, 28, 35, and 42 days after yolk sac collapse (DAYC) were immersed in formalin-killed *Streptococcus agalactiae* vaccine (FKV-SA). A specific IgM was first detected via ELISA in the 21 DAYC larvae (0.108 g) at 336 h after vaccination (hav), whereas in the 28–42 DAYC larvae (0.330–0.580 g), the specific IgM could be initially detected at 24 hav. qRT–PCR analysis of the *TCRβ*, *CD4*, *MHCIIα*, *IgHM*, *IgHT*, and *IgHD* genes in 21–42 DAYC larvae immunized with the FKV-SA immersion route for 24, 168, and 336 hav revealed that the levels of most immune-related genes were significantly higher in the vaccinated larvae at all DAYCs than in the control larvae (*p <* 0.05) at 336 hav. Immunohistochemistry demonstrated stronger IgM signals in the gills, head kidney, and intestine tissues at 21, 28, and 35 DAYC in all vaccinated larvae compared with the control. Interestingly, at all DAYCs, FKV-SA larvae exhibited significantly higher survival rates and an increased relative percent survival (RPS) than the control after challenge with viable *S. agalactiae*, particularly in larvae that were immunized with FKV-SA at 168 and 336 hav (*p* < 0.05).

## 1. Introduction

*Oreochromis niloticus*, classified as a vertebrate, is a species of freshwater cichlid commonly known as Nile tilapia. As in many countries worldwide, it is an economically important freshwater fish with high aquaculture value and productivity in Thailand [1]. Due to the increasing demand for aquatic food as a protein source, the production capacity per unit area has increased with intensive aquaculture to provide more food of high quality. This results in large amounts of leftover feed and excreta, causing the accumulation of organic and inorganic substances in the culture ponds, leading to sudden changes in culture pond conditions, especially chemical, biological, and physical factors associated with water quality. These fluctuations lead to stress, causing the fish to become weakened and more susceptible to various pathogens. Among these pathogens, *Streptococcus* spp., especially *S. agalactiae* and *S. iniae*, can cause streptococcosis, which is more severe in summer, when the temperature is usually high. The clinical signs of this disease include exophthalmia, skin hemorrhage, erratic swimming, and high mortality. The disease is severely damaging Thailand’s tilapia aquaculture industry, as also observed in other countries in temperate and tropical regions [1,2,3,4].

All bacteria in the genus *Streptococcus* belong to Streptococcaceae. These Gram-positive bacteria are classified as lactic acid bacteria, are pathogenic, and can cause high mortality rates of up to 75%, with an economic impact on the aquaculture of tilapia and other fish species [5,6]. Generally, antibiotics and chemicals are used to control bacterial diseases. However, the application of these agents has several adverse effects, such as the contamination of the environment and fish products, which further affects beneficial microorganisms and the health of consumers. Additionally, the drug resistance of pathogenic bacteria is frequently observed and eventually leads to failure in the control of these pathogens in fish culture systems [7,8].

To address these concerns, vaccines are used as an effective strategy for disease control. The development of vaccines for fish aquaculture has advanced, and many commercial vaccines have been developed for use in several of the most economically important fish species at the commercial scale [9,10,11], including for Nile tilapia [12,13,14]. There remain some critical limitations associated with this strategy, including the impracticality of the vaccination route. Among the vaccine administration routes (injection, oral, and immersion), injection is the most effective based on immune responses and protective efficacy [15,16], while oral and immersion vaccinations are more practical but less effective than injection. Therefore, oral administration and immersion vaccination are the major practices for increasing vaccine efficacy, especially immersion vaccination. Compared with other methods, the immersion route has many advantages, including being less labor-intensive, being able to minimize stress, and being administrable to fish of various sizes, particularly small fish or larvae, at low amounts [17,18].

Based on current information, immersion vaccination in the early stages of fish development is not as successful as other methods. One of the biggest challenges is understanding the optimal stages for the prompt exploration of maximal immune responses, as little is known about this, especially in Nile tilapia. Therefore, we aimed to investigate the early immune responses of Nile tilapia larvae following immersion vaccination with inactivated *S. agalactiae*. Our investigation encompasses an assessment of specific IgM antibody responses, an analysis of immune gene expression, and an evaluation of vaccine efficacy through challenge tests. The basic knowledge arising from this study will be an important part of creating an effective strategy to prevent severe losses caused by various disease outbreaks in Nile tilapia. These results will also be essential for the Nile tilapia aquaculture industry, especially for enhancing production and reducing costs and the negative side effects from antibiotic and chemical applications, which are crucial for sustainable Nile tilapia aquaculture.

## 2. Materials and Methods

### 2.1. Experimental Fish and Husbandry

A total of 35 male, apparently healthy (tested negative for pathogens) Nile tilapia (*Oreochromis niloticus*), approximately 6 months of age, weighing 200 ± 25 g, and with a length of 20.6 ± 1.5 cm, and 96 females, weighing 200 ± 25 g and with a length of 18.8 ± 1.3 cm, were bred in a 5 × 10 m^2^ cement tank containing freshwater with a full aeration system at the Fisheries Research Station, Kamphaengsaen campus, Nakornprathom Province, Thailand. Microbiological techniques important for disease diagnosis were employed weekly to monitor fish health, especially the presence of external parasites and bacterial infection [3,4]. During spawning, eggs were collected from female fish and kept in a flow-through hatchery system until they reached the fifth stage (yolk sac collapse (YSC) larvae). Then, 7000 larvae each from fish at the same stage were further moved to a larval nursery system in 23,000 L tanks. Fish larvae were acclimatized in a 3000 L tank with aerated freshwater at a temperature of 28 ± 2 °C, dissolved oxygen (DO) level of 5 ± 1.5 mg/L, and pH 7.8 ± 1.0. Water quality was monitored, and the water was replaced on alternate days. The fish were fed commercial larval feed (Hi-Grade 9006T, Charoen Pokphand, Thailand) with an approximate crude protein content of 42% at 10% of the body weight 3 times a day for 42 days (42 DAYC).

A schematic overview of all experiments in the current study is illustrated in the Appendix A. The experimental procedures and the use of animals were approved according to the guidelines of the Animal Welfare Committee of Kasetsart University (ACKU65-FIS-016).

### 2.2. Bacterial Cultivation and Preparation of the Formalin-Killed S. agalactiae Vaccine (FKV-SA)

#### 2.2.1. Bacterial Preparation

*Streptococcus agalactiae* bacterium serotype III was previously isolated from diseased Nile tilapia and identified at the Center of Excellence in Aquatic Animal Health Management, Department of Aquaculture, Faculty of Fisheries, Kasetsart University, Bangkok, Thailand. A single colony of the target strain was inoculated into tryptic soy broth (TSB; Becton, MD, USA) and incubated overnight at 37 °C for 24 h with shaking at 120 rpm. Pellets were obtained via centrifugation at 5000× *g* for 12 min and washed twice with 0.85% NaCl, and then, the supernatant was discarded. Bacterial concentrations were monitored using a spectrophotometer (Milton Roy, PA, USA) until the optical density at 600 nm reached 1.0 to obtain a final bacterial cell concentration of approximately 1 × 10^9^ CFU/mL. This culture was further diluted for use in the other experiments, and the number of bacteria was always confirmed by spreading on tryptic soy agar (TSA).

#### 2.2.2. Formalin-Killed *S. agalactiae* Vaccine (FKV-SA) Preparation

The above-prepared bacterial cells were inactivated in 1.0% (*v*/*v*) formalin solution and incubated overnight at 4 °C. Bacterial inactivation was verified by spreading 100 µL on TSA (Becton, Dickinson and Company, USA) and incubation at 37 °C for 16–24 h. When no colonies were observed, the inactivated bacterial suspension was prepared as described above, and the bacterial pellet was washed twice with 0.85% NaCl and centrifuged at 5000× *g* and 4 °C for 12 min. The dead cell density was measured for the absorbance using the above methods to obtain a final bacterial cell concentration of approximately 1 × 10^9^ CFU/mL, and the cells were maintained at 4 °C until use.

### 2.3. Efficacy of the S. agalactiae Immersion Vaccine (FKV-SA) in Nile Tilapia Larvae

#### 2.3.1. Experimental Design

Using a completely randomized design (CRD), a 2 × 5 factorial experiment with two replicates was designed with two factors (vaccination and days after vaccination). Seven different Nile tilapia larva groups (different DAYCs) at 1, 7, 14, 21, 28, 35, and 42 DAYC (as described in Section 2.1) were used in this study. Nile tilapia larvae were randomly divided from the storage tanks into four 500 L tanks, each group containing 500 fish (250 fish/tank). Tanks 1 and 2 were assigned as control groups 1 and 2, and tanks 3 and 4 were assigned as vaccinated groups 1 and 2 (FKV-SA1 and 2). A total of 28 experimental tanks (2 tanks for control × 2 tanks for vaccinated groups × 7 periods) were set up. The fish at different larval stages were weighed and scaled for growth parameter determination.

#### 2.3.2. Immersion Vaccination

The vaccine, prepared as described in Section 2.2.2, was diluted to a concentration of 1 × 10^7^ CFU/mL. At each fish larval stage, 4 20 L tanks were prepared. Tanks 1 and 2 were the control group immersed in 0.85% NaCl, and tanks 3 and 4 were immersed in formalin-killed *S. agalactiae* vaccine (FKV-SA) at a concentration of 1 × 10^7^ CFU/mL for 2 h. All tanks were continuously aerated during immunization, and water temperatures were maintained at 28 ± 2 °C. After vaccination, fish were returned to their nursery tanks.

#### 2.3.3. Sample Collection

The whole bodies of 18 tilapia larvae in the vaccinated and unvaccinated groups (9 fish per tank) were randomly collected at 1, 7, 14, 21, 28, 35, and 42 DAYC at 0, 6, 24, 168, and 336 h after vaccination (hav). Six fish were stored in TRIzol™ reagent (Thermo Fisher Scientific, MA, USA) at −80 °C for gene expression studies for a short period, and the other 6 whole fish samples were kept at −20 °C for an assessment of total antibody amounts via ELISA. The 6 remaining whole tilapia larvae were preserved in 10% buffer formalin solution at room temperature for immunohistochemical analysis.

#### 2.3.4. Measurement of Fish Weight and Size Prior to and Post-Vaccination

Two growth parameters in each experimental group were evaluated: body weight and total length. Six fish from each control and treatment group were randomly selected on each DAYC at 1, 7, 14, 21, 28, 35, and 42 h and at 0, 6, 24, 168, and 336 h after inoculation to measure weight and length.

#### 2.3.5. Quantification of Specific IgMs against FKV-SA via ELISA

Fish larvae from the previous section were separately extracted for protein analysis using previously described methods [19]. Briefly, the whole body of Nile tilapia larvae was individually collected into 1.5 mL Eppendorf tubes and preserved in 1 mL of a protein extraction buffer containing a protease inhibitor cocktail (HiMedia, Mumbai, India). The larvae samples were homogenized using pellet pestles under a stable temperature condition at 4 °C and centrifuged at 2000–2500 rpm for 5 min; the supernatant was removed to a new 1.5 mL Eppendorf tube and stored at −80 °C. The ELISA microplates were first coated with carbonate–bicarbonate coating buffer, pH 9.6, for 1 h, and the supernatant was discarded and washed three times with 1× low-salt wash buffer (LSWB). One hundred microliters of FKV-SA at 1 × 10^9^ CFU/mL in PBS was added to the wells of the plates. The plates were incubated overnight at 4 °C, 50 μL of 0.05% glutaraldehyde was added, and the plate was allowed to stand for 20 min at room temperature (RT). Then, the glutaraldehyde was discarded, and the wells were washed three times with 1× LSWB. The microplate wells were further blocked with 100 μL of 1% bovine serum albumin (BSA) blocking buffer (Sigma A9647) for 2 h at RT and then washed five times with 1× high-salt wash buffer (HSWB). Then, 100 μL of the proteins extracted from sampled fish was added to the wells, and the plate was incubated overnight at 4 °C. After washing the plate five times with HSWB, an anti-tilapia IgM monoclonal antibody (1:200 *v*/*v*) (Marine Leader, Bangkok, Thailand) was added for 2 h. The plates were washed five times with HSWB and incubated for 5 min on the last wash. Then, 100 μL of conjugated anti-mouse IgG-HRP (Sigma Aldrich, USA) diluted at 1:3000 *v*/*v* was added, and the plate was incubated for 1 h at RT. The plates were washed five times and incubated for 5 min on the last wash. One hundred microliters of TMB (3′,3,5′,5-tetramethylbenzidine dihydrochloride) was added to each well, and the plates were incubated at RT for 10 min in the dark. The reaction was stopped with 100 μL of H_2_SO_4_ (Sigma–Aldrich 320501). The absorbance of the reaction was measured with an iMark™ Microplate Absorbance Reader (Bio-Rad iMarkTM microplate reader). The serum antibody levels specific to the FKV-SA antigen of fish larvae at each DAYC and the day after vaccination were determined using the absorbance values at 450 nm.

#### 2.3.6. Determination of Immune-Related Gene Expression in Tilapia Larvae after Immersion Vaccination

The whole bodies of six fish from each treatment were randomly sampled from each replicate at 21, 28, 35, and 42 days at 24, 168, and 336 h post-immersion vaccination. The whole bodies of the larvae were placed in 1.0 mL of TRIzol™ reagent (Thermo Fisher Scientific, USA) and immediately homogenized with an automatic tissue extractor (MP, CA, USA). Total RNA was further extracted according to the manufacturer’s instructions (Thermo Fisher Scientific, USA). Total RNA was isolated from the tissues, and the RNA concentration and purity were determined using a Nanodrop spectrophotometer (Bio-Rad Laboratories, Hercules, CA, USA). One microliter of 1000 ng/μL total RNA was then used as a template to synthesize first-strand cDNA using the ReverTra Ace^®^ qPCR RT Master Mix with gDNA Remover Kit (TOYOBO, Osaka, Japan). The first-strand cDNA synthesis product was stored at −80 °C for further analysis.

The expression of the immune-related genes *TCRβ*, *CD4*, *MHCIIα*, *IgM*, *IgT*, and *IgD* was examined as described by Bunnoy et al. (2023) [19]. The specific primers are shown in Table 1. The qRT–PCR assay was performed using Brilliant III Ultra-Fast SYBR^®^ Green (Agilent, CA, USA) in an AriaMx Real-Time PCR system (MY18135206). qPCR was performed in a total volume of 20 μL containing 10 μL of 2 × SYBR Green QPCR Master Mix, 2 μL of 0.5 mM forward and reverse primers, and 1 μL of cDNA template, adjusting the reaction mixture to a final volume of 20 μL with distilled water, and the thermal cycler program was as follows: 95 °C for 5 min; 40 cycles of 95 °C for 30 s, 55 °C for 30 s, and 72 °C for 90 s; and a final extension at 72 °C for 10 min (as previously described [20]). All reactions were performed in triplicate. The expression levels in each sample were described based on the relative fold-change compared to the expression of *β-actin* calculated using the 2^−△△CT^ method according to the protocol by Livak and Schmittgen (2001) [21].

#### 2.3.7. Immunohistochemical Analysis

Whole-body samples of tilapia larvae were randomly collected from six fish from each replicate at days 21, 28, and 35 and at 0, 6, 24, 168, and 336 h after vaccination for immunohistochemical analyses. The samples were preserved in a 10% buffered formalin fixative solution for 24 h. They were then dehydrated in a graded ethanol series and xylene before embedding in paraffin. Sections were mounted on glass slides and dried for 24 h at 37 °C. The slides were deparaffinized in xylene and rehydrated in a graded ethanol (EtOH) series, initially using absolute ethanol for 5 min (twice), followed by 80% EtOH for 5 min and 70% ethanol for 5 min. The slides were then washed with distilled water for 5 min. Then, the slides were pretreated for 30 min with 3% H_2_O_2_ in 70% ethanol in the dark at RT and then washed with PBS. The slides were incubated with 1% glycine for 30 min at RT. Antigens were retrieved by breaking the crosslinks between the antigens and fixative or paraffin, boiling in citrate buffer, placing the slides for 10 min on a hot plate and naturally cooling at RT. The slides were washed with 1× PBST for 5 min (twice) and permeabilized with 0.4% Triton X-100 in 1× PBS for 5 min (twice). The slides were further blocked with 5% BSA (in 1× PBS) for 30 min in a humid box at RT and washed with 1× PBST for 5 min (twice). The slides were then incubated at 37 °C for 3 h with primary antibody (anti-Nile tilapia IgM, as mentioned in Section 2.3.5) diluted in PBS (1:100). After washing, the slides were incubated with GAM-HRP (secondary antibody, 1:1000) in 1× PBS and incubated at 37 °C for 2 h with 1× PBST for 5 min (3 times). Then, the bound antigen was visualized by incubating the slides in DAB staining solution in the dark for 15 min, counterstaining with Meyer hematoxylin, and mounting in Permount™ mounting medium (PMM). Images of the sections were collected using a compound light microscope (Olympus, MA, USA). Areas showing positive DAB signals were stained brown, and nuclei were stained blue. The light brown or deep brown areas on the observed tissues were duly recorded as Nile tilapia IgM-positive detection.

### 2.4. Effects of Vaccination in Different Nile Tilapia Larval Stages (DAYCs 21, 28, 35, and 42) on Disease Resistance against S. agalactiae

#### 2.4.1. Assessment of the Time-Dependent Toxicity of the Lethal Concentration (LC_50_) of *S. agalactiae* in Nile Tilapia

Fish prepared as described in Section 2.1 were used. The fish were divided into 6 experimental groups with 3 replicates (30 fish/replicates) as follows: group 1, which was the control group, was inoculated with PBS, and groups 2, 3, 4, 5, and 6 were inoculated with viable *S. agalactiae* at concentrations of 1 × 10^4^, 1 × 10^5^, 1 × 10^6^, 1 × 10^7^, and 1 × 10^8^ CFU/mL, respectively. The fish were exposed to the pathogens via immersion for 2 h and transferred to their tanks. The mortality rate was observed and recorded daily. The survival rate was then calculated for the median lethal concentration (LC_50_) according to the methods of Finney (1971) [24] and Litchfield and Wilcoxon (1949) [25].

#### 2.4.2. Vaccination of Fish Larvae

Unvaccinated fish larvae at 21–42 DAYCs from Section 2.1 were used and 270 larvae each for control and treatment groups at each DAYC were stocked in a 250 L glass tank with the above conditions. The experiment was further divided into 4 experimental groups according to the age of the tilapia larvae: Groups 1–4 were at DAYC 21, 28, 35, and 42, respectively. At each DAYC, in tanks 1–3 (250 L glass tanks), which served as controls, 30 larvae were maintained, with 3 repetitions. Tanks 4–6 were simultaneously prepared with the same conditions as those used for the vaccinated groups. For vaccination, fish in tanks 1, 2, and 3 were collected using a round scope net and immersed in PBS (pH 7.4) for 2 h, and fish in tanks 4, 5, and 6 were immersed in FKV-SA at a concentration of 1 × 10^7^ CFU/mL for 2 h. Then, they were transferred back to the tanks in which they were raised (Appendix A).

#### 2.4.3. Challenge Test

At 24, 168, and 336 h after immunization, thirty fish larvae from each tank of both control and vaccinated fish at each DAYC were then immersed in an *S. agalactiae* bacterial solution at the LC_50_ value of 3.34 × 10^5^ CFU/mL for 2 h and transferred to a 250 L container with the previous conditions. Then, fish mortality was recorded daily for 8 days after immunization. The survival rate and relative percent survival (RPS) were calculated as follows:

Survival rate (%) = (number of fish remaining/total number of fish) × 100

RPS = 1 − (mortality rate of vaccinated fish/mortality rate of control fish) × 100.

### 2.5. Statistical Analysis

IgM levels, immune-related gene expression, mortality, and growth performance were analyzed via two-way analysis of variance (ANOVA) and Duncan’s multiple comparisons test to determine differences between groups. The assumption of normal distributions and homogeneity of variance were carefully checked before analysis. The results are expressed as the means ± SDs. The interactions between the determined factors were further analyzed. All statistical analyses were performed using IBM SPSS Statistics version 25 and GraphPad Prism 8.0.2. Survival analysis of Nile tilapia larvae at different vaccination ages after challenge with *S. agalactiae* was performed using the Kaplan–Meier method. Significant differences in survival rates between the control and vaccinated groups in each period in each DAYC group were statistically analyzed using Student’s t tests. The level of statistical significance between the control and experimental groups for all parameters was indicated as *p* < 0.05.

## 3. Results

### 3.1. Determination of the Levels of Serum IgM Specific to S. agalactiae via ELISA

An analysis of the inactivated *S. agalactiae* vaccine (FKV-SA) via immersion at time intervals of 0, 6, 24, 168, and 336 hav showed no significant differences in the measured specific IgM levels against *S. agalactiae* at 1, 7, and 14 DAYC compared with the nonvaccinated control group (*p* > 0.05) (Figure 1A–C). However, in Nile tilapia larvae immunized with FKV-SA at 21 DAYC, a statistically significant increase in IgM levels was observed at 336 hav compared with the nonvaccinated control group (*p* < 0.05) (Figure 1D). In addition, a significant increase in IgM levels specific to *S. agalactiae* was observed in Nile tilapia larvae at 28, 35, and 42 DAYC after vaccine immunization, especially at 24, 168, and 336 hav, respectively, compared with the nonvaccinated control group (*p* < 0.05) (Figure 1E–G). Furthermore, IgM levels increased in both the vaccinated and unvaccinated treatments during every experimental period when overall vaccination periods were considered. However, no interaction between periods and vaccination treatments was observed throughout the vaccination periods (*p* > 0.05).

**Figure 1 vaccines-11-01753-f001:**
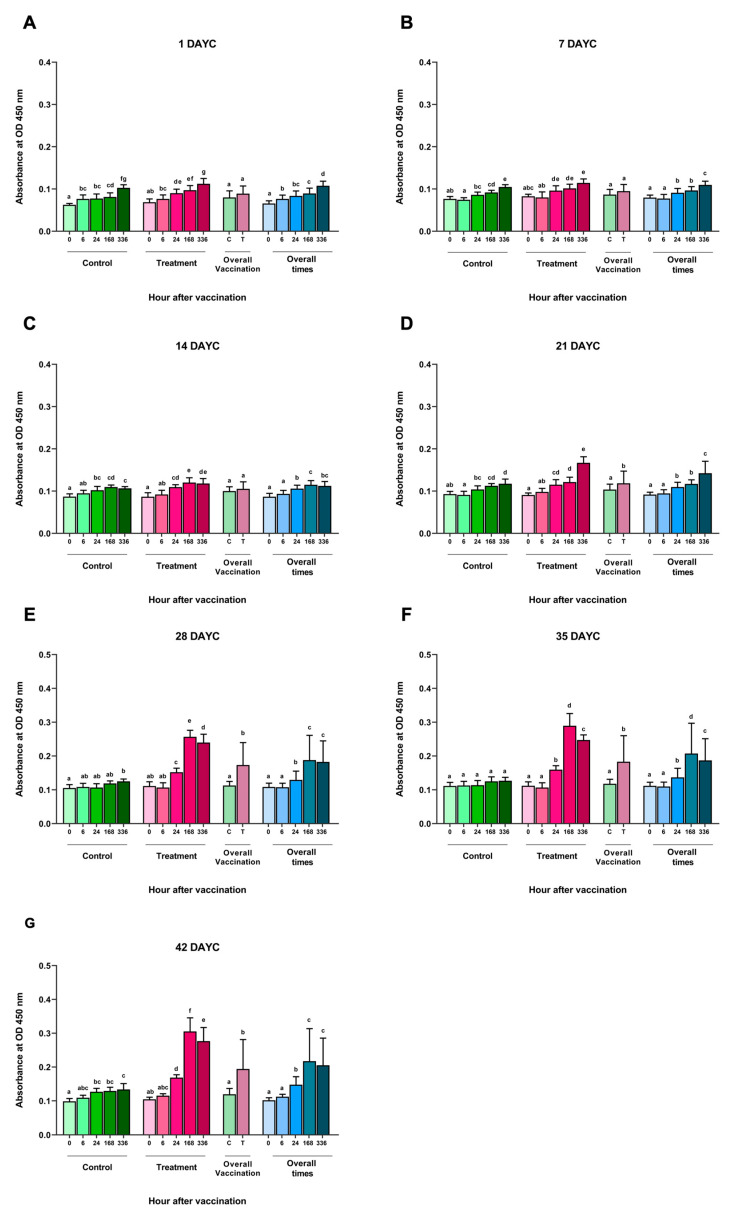
*S. agalactiae*-specific IgM levels in FKV-SA-treated Nile tilapia. The IgM levels were determined via ELISA at 1 (**A**), 7 (**B**), 14 (**C**), 21 (**D**), 28 (**E**), 35 (**F**), and 42 (**G**) DAYC. Different letters on each bar denote significant differences (*p* < 0.05).

**Figure 2 vaccines-11-01753-f002:**
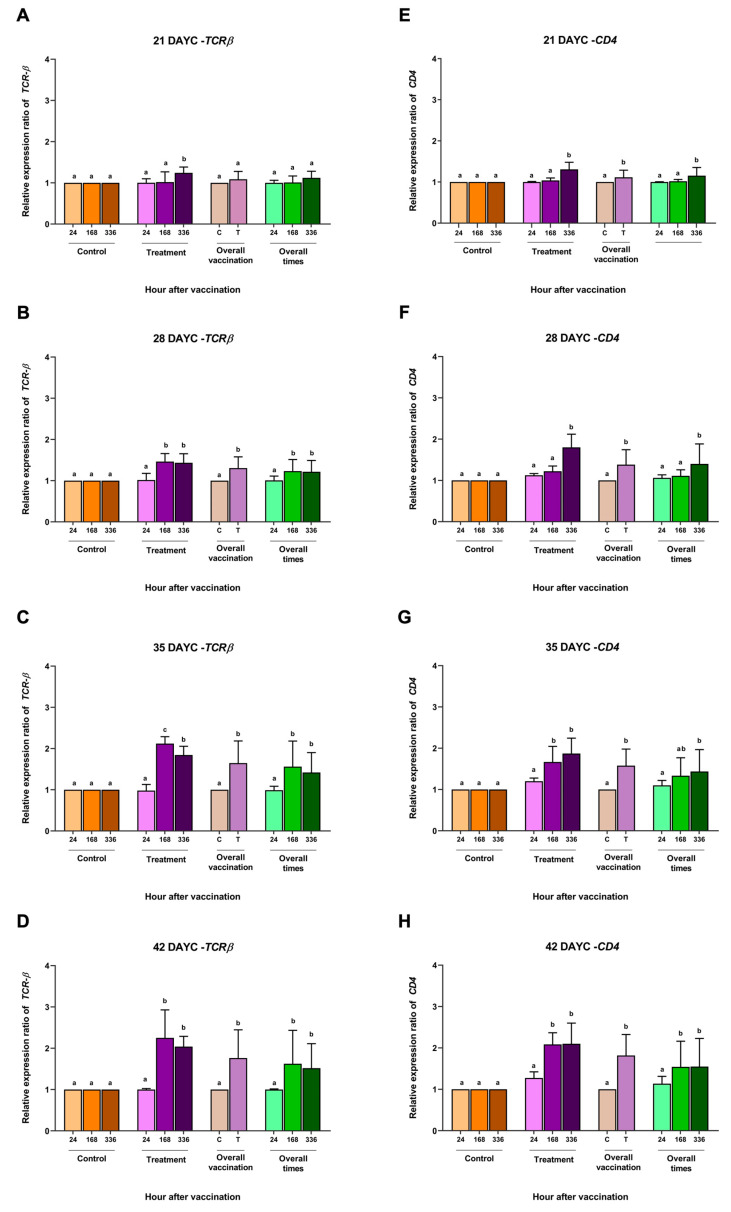
*TCRβ* (**A**–**D**) and *CD4* (**E**–**H**) gene expression at 21, 28, 35, and 42 DAYC. Expression levels were measured via qRT–PCR. Different letters on each bar denote significant differences (*p* < 0.05).

### 3.2. Immune-Related Gene Expression

Based on the IgM levels determined in Section 3.1, the relative gene expression of the immune-related genes *IgM*, *IgT*, *IgD*, *MHCIIα*, *TCRβ*, and *CD4* in the whole body of Nile tilapia larvae was determined at 21, 28, 35, and 42 DAYC and at 24, 168 and 336 hav. The relative expression of the TCRβ gene in fish larvae at 21 DAYC initially exhibited differences between the vaccinated and nonvaccinated groups at 336 hav. However, fish larvae at 28–42 DAYC and immunized with FKV-SA were found to express the *TCRβ* gene at significantly higher levels than control fish (Figure 2A–D). This was consistent with the overall vaccination result, wherein the expression of the *TCRβ* gene was first found to be significantly different in 28 DAYC larvae. When the effects of overall vaccination times were considered, significantly higher levels of the *TCRβ* gene from 28–42 DAYC larvae at 168 and 336 hav were observed (Figure 2B–D).

The overall *CD4* gene expression in fish larvae immunized with FKV-SA at 21–42 DAYC significantly differed from that in control larvae (*p* < 0.05) (Figure 2E–H). In the overall vaccination period, 21, 28, and 35 DAYC larvae showed significant differences in *CD4* gene expression at 336 hav (*p* < 0.05) (Figure 2E–G), while 42 DAYC larvae showed significantly higher *CD4* gene expression at 168 and 336 hav (*p* < 0.05) (Figure 2H).

The relative *MHCIIα* gene expression in larvae immunized with FKV-SA 21–42 DAYC showed similarly significant differences compared with the control group at 336 h after vaccination (*p* < 0.05) (Figure 3A–D). Similarly, significant differences in overall vaccination were observed early in 21 DAYC larvae (Figure 3A). Similar results were obtained for the overall vaccination period, with 28–42 DAYC larvae showing a significant increase only in the group that received the vaccine at 336 hav (*p* < 0.05) (Figure 3A–D).

The relative *IgM* gene expression differed significantly only in the 21 DAYC larvae immunized with FKV-SA (*p* < 0.05) (Figure 3E), while the 28, 35, and 42 DAYC larvae immunized with FKV-SA showed the upregulation of IgM gene expression at 168 and 336 hav (*p* < 0.05) (Figure 3F–H). Interestingly, all the larvae immunized with FKV-SA exhibited a significant difference compared to the 21–42 DAYC larvae (*p* < 0.05) (Figure 3E–H). Overall, the 21 DAYC larvae showed a significant difference at 336 hav, while the 28, 35, and 42 DAYC larvae showed upregulation at 168 and 336 hav (*p* < 0.05) (Figure 3F–H).

For *IgT* gene expression, at 21 DAYC, only the larvae immunized with FKV-SA showed a significant increase at 336 hav (*p* < 0.05) (Figure 4A). The vaccinated fish larvae at 28 and 35 DAYC showed significant differences compared with the other groups at 168 and 336 hav (*p* < 0.05) (Figure 4B,C). However, at 42 DAYC, all immunized fish larvae in all vaccination periods showed significant differences compared to the control larvae (*p* < 0.05) (Figure 4D). Additionally, overall vaccination of the control and vaccinated groups was observed at all DAYCs (*p* < 0.05) (Figure 4A–C). Overall, significant differences were observed in 21, 28, and 42 DAYC larvae at 168 and 336 hav (*p* < 0.05) (Figure 4A–C). Finally, in terms of relative *IgD* gene expression, no significant differences were observed among treatments, overall vaccination, and overall times in the 21 and 28 DAYC larvae (*p* > 0.05) (Figure 4E,F). Significant differences in *IgD* gene expression were observed in the 35 and 42 DAYC larvae at 168 and 336 hav (*p* < 0.05) (Figure 4G,H). Significant differences in overall vaccination were observed between the control and vaccinated groups in both 35 and 42 DAYC larvae (*p* < 0.05) (Figure 4G,H), while the overall vaccination periods were significant in both 35 and 42 DAYC larvae at168 and 336 hav (*p* < 0.05) (Figure 4G,H).

No interaction effect of periods and vaccination treatments on the expression levels of the genes examined was observed throughout the vaccination periods (*p* > 0.05).

### 3.3. IgM Distribution in Nile Tilapia Larvae after Immersion Vaccination

The distribution of IgM in the gills, head kidneys, and intestines of Nile tilapia larvae at 21, 28, and 35 DAYC and 336 hav is shown in Figure 5. At 21 DAYC, the distribution of IgM was observed in the tissues of head kidneys and intestines of vaccinated fish (Figure 5G,M) compared with the nonvaccinated fish (Figure 5J,P). In the gills, a very weak signal was observed in the gill lamellae of both groups (Figure 5A,D). At 28 and 35 DAYC, fish larvae exposed to FKV-SA exhibited strong IgM signals in the head kidney (Figure 5H,I) and intestines (Figure 5N,O), which were more obvious than those in control fish (Figure 5K,L,Q,R, respectively). In the case of the gills, at 28 and 35 DAYC, higher expression of the IgM protein was noted, with higher levels in fish larvae immersed in FKV-SA (Figure 5B,C) than in the control groups (Figure 5E,F).

**Figure 3 vaccines-11-01753-f003:**
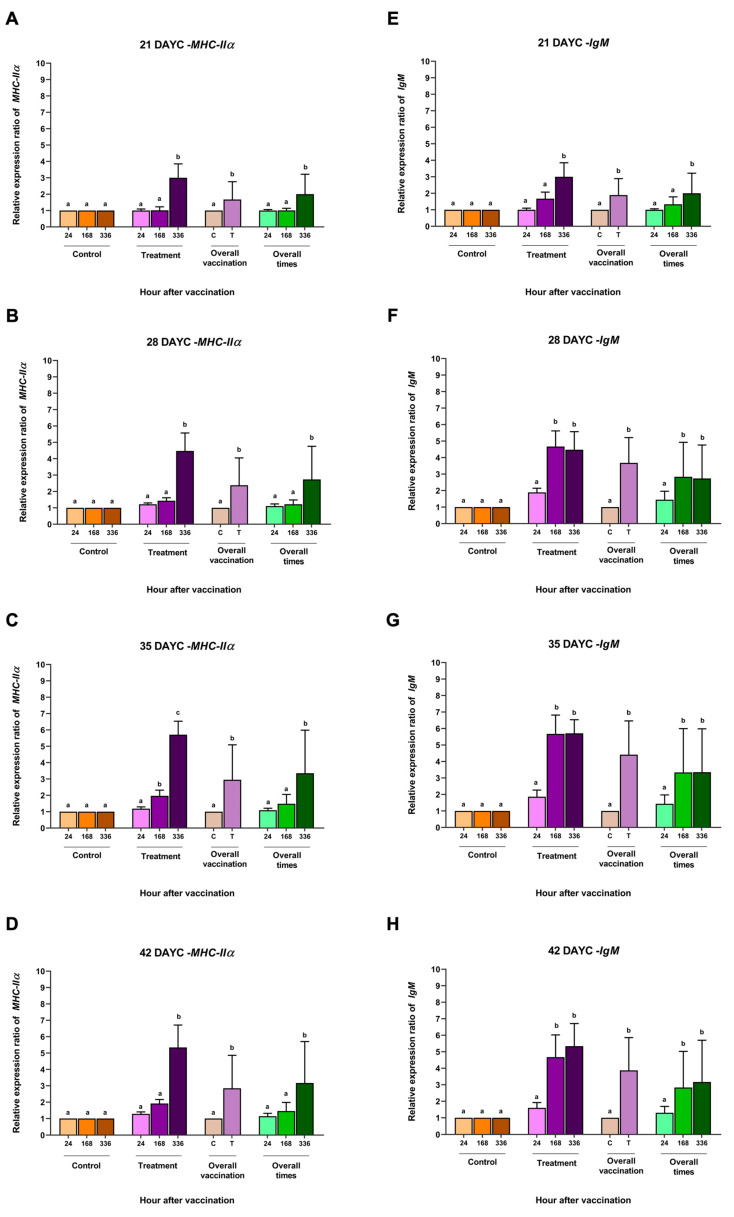
Gene expression analysis of *MHCIIα* (**A**–**D**) and *IgM* (**E**–**H**) at 21, 28, 35 and 42 DAYC. Expression levels were measured via qRT–PCR. Different letters on each bar denote significant differences (*p* < 0.05).

**Figure 4 vaccines-11-01753-f004:**
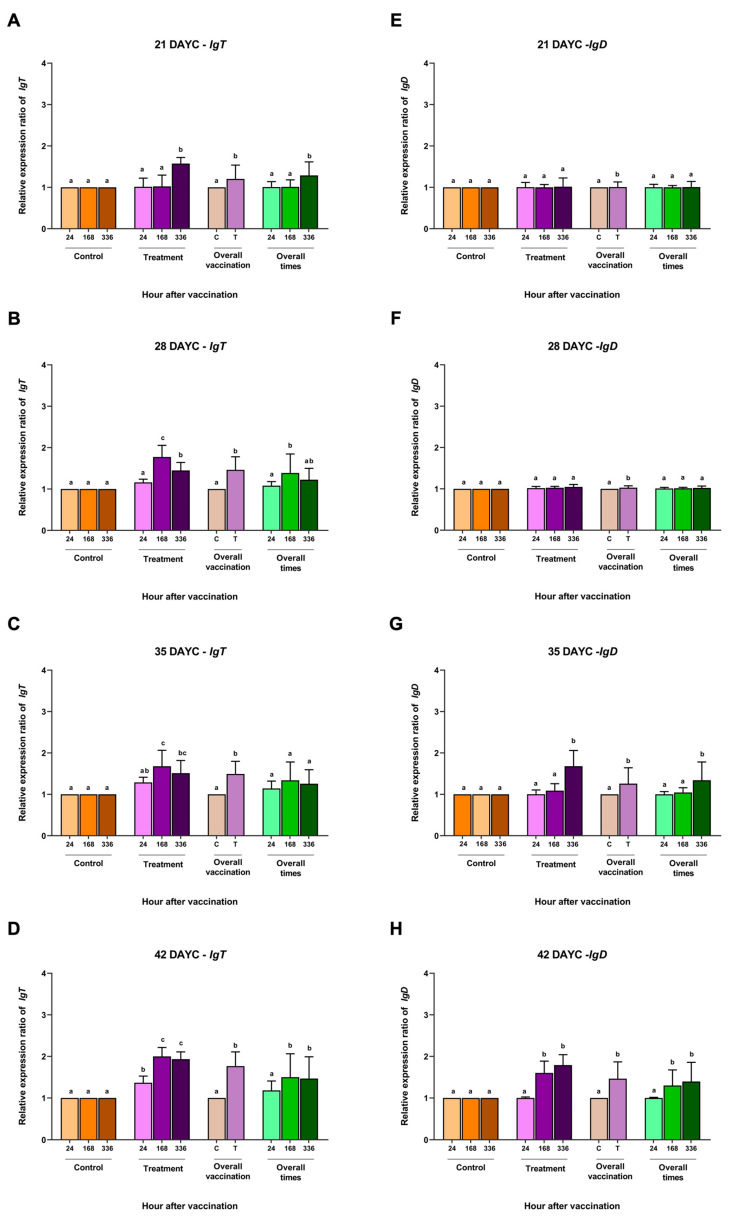
*IgT* (**A**–**D**) and *IgD* (**E**–**H**) gene expression at 21, 28, 35 and 42 DAYC. Expression levels were measured via qRT–PCR. Different letters on the bars denote significant differences (*p* < 0.05).

**Figure 5 vaccines-11-01753-f005:**
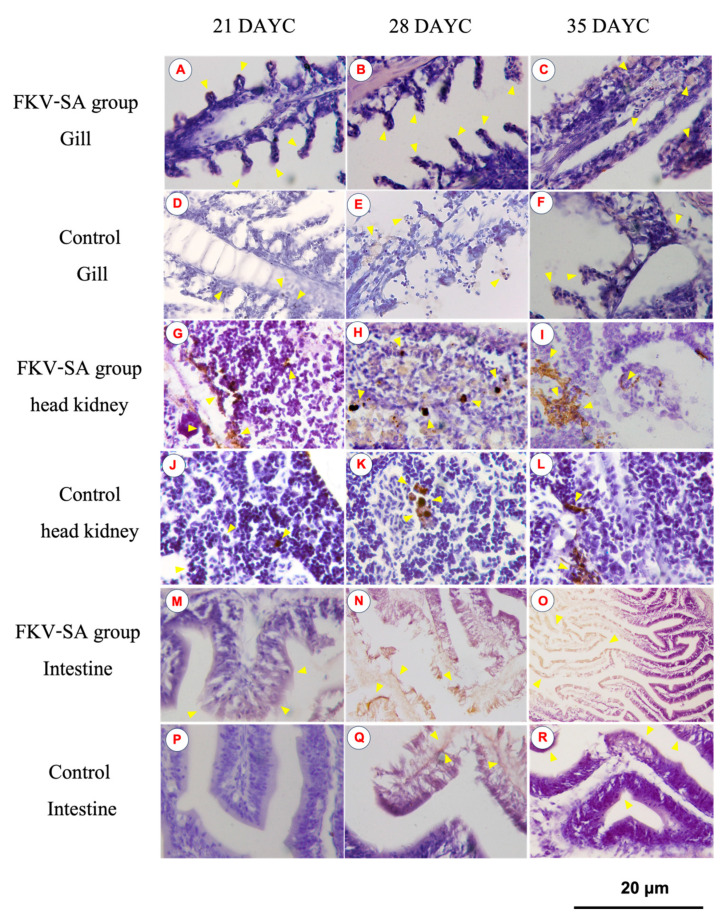
Distribution of IgM in Nile tilapia larvae at 336 h after immersion in FKV-SA based on immunohistochemistry of immunized fish gills of the FKV-SA group (**A**–**C**) and control group (**D**–**F**), head kidneys of the FKV-SA group (**G**–**I**) and control group (**J**–**L**), and intestines of the FKV-SA group (**M**–**O**) and control group (**P**–**R**) at 21, 28, and 35 DAYC. Yellow arrows indicate IgM-positive signal areas.

### 3.4. Weight and Length Measurements

The effects of FKV-SA on the weight and total length of Nile tilapia larvae were investigated during vaccination trials. At 336 hav, the weights of fish larvae in the control groups at 1, 7, 14, 21, 28, 35, and 42 DAYC ranged from 0.036 ± 0.022 to 0.750 ± 0.123 g. In contrast, in the vaccinated groups, the weights of fish larvae in the control groups at 1, 7, 14, 21, 28, 35, and 42 DAYC ranged from 0.038 ± 0.027 to 0.821 ± 0.190 g. The lengths of fish larvae in the control groups at 1, 7, 14, 21, 28, 35, and 42 DAYC were 1.10 ± 0.6 to 1.94 ± 0.2 cm. Furthermore, the values for the fish larvae in the vaccinated groups at 1, 7, 14, 21, 28, 35, and 42 DAYC were 1.30 ± 0.1 to 2.18 ± 0.2 cm. However, the growth indices showed no statistically significant changes (Appendix A).

### 3.5. Lethal Concentration (LC_50_) of S. agalactiae

On day 8, the cumulative percent mortality of Nile tilapia larvae at concentrations of 1 × 10^4^, 1 × 10^5^, 1 × 10^6^, 1 × 10^7^, and 1 × 10^8^ CFU/mL was 31.11 ± 3.33, 35.56 ± 6.94, 47.78 ± 5.09, 74.44 ± 8.39, and 82.22 ± 3.33%, respectively (Figure 6). Based on the probit analysis, the LC_50_ of *S. agalactiae* in Nile tilapia larvae was 3.44 ± 1.26 × 10^5^ CFU/mL. This concentration was further used for the challenge test.

### 3.6. Survival Analysis and RPS of Nile Tilapia Challenged with S. agalactiae

In this experiment, 21, 28, 35, and 42 DAYC Nile tilapia larvae immunized with FKV-SA for 24, 168, and 336 hav were challenged with viable *S. agalactiae* to test the efficacy of the vaccine with various exposure times. The survival rates of vaccinated Nile tilapia larvae after challenge with *S. agalactiae* are shown in Figure 7A–H. At 21 DAYC, the survival of larvae immunized with only FKV-SA for 336 hav significantly differed from that of control fish larvae (*p* < 0.05), with values of 58.67 ± 2.31 and 42.00 ± 3.61%, respectively (Figure 7A). In 28 DAYC larvae, significant differences in survival between vaccinated and unvaccinated groups were observed at 168 hav (66.00 ± 4.58 and 40.61 ± 2.08%) and 336 hav (62.00 ± 2.00 and 44.00 ± 2.65%) (*p* < 0.05) (Figure 7B). Similar results were observed in the 35 and 42 DAYC larvae. At 35 DAYC, vaccinated fish larvae showed significant differences at 168 and 336 hav, with survival rates of 65.33 ± 2.31 and 62.00 ± 5.57%, respectively, compared with the control (40.67 ± 1.53 and 44.67 ± 2.08%, respectively) (*p* < 0.05) (Figure 7C). Similarly, at 42 DAYC, the vaccinated fish larvae showed significant differences at 168 and 336 hav, with survival rates of 68.67 ± 3.79 and 62.67 ± 2.52%, respectively, compared with the control (42.00 ± 4.00 and 46.00 ± 2.65%, respectively) (*p* < 0.05) (Figure 7D). Moreover, the RPSs of fish larvae immunized with FKV-SA at various exposure times were also calculated. Larvae at 21 and 28 DAYC and at 24, 168, and 336 hav showed no significant difference among exposure times (*p* > 0.05) (Figure 7E,F). However, at 35 and 42 DAYC, fish larvae immunized with FKV-SA for 168 h showed significantly higher differences in RPSs than those immunized for 24 and 336 h, showing the highest RPSs of 41.53 ± 7.61 and 45.54 ± 13.44%, respectively (*p* < 0.05) (Figure 7G,H).

## 4. Discussion

Streptococcosis caused by *S. agalactiae* severely affects Nile tilapia aquaculture worldwide [26,27,28,29]. Vaccines are one of the first measures selected to address this problem. Various studies have reported the development of effective vaccines against *S. agalactiae* infection in Nile tilapia [13,30,31,32,33]. The results obtained for most developed vaccines showed that they were very effective in the laboratory [34,35,36] and on the field-trial scale [37,38,39], but most of those studies relied on injection methods [40,41,42]. This vaccination route was subsequently found to be impractical and unacceptable for field application in many fish species. Therefore, a more practical vaccination is needed. Oral administration and immersion vaccination are the most important vaccination routes of interest. Immersion vaccination has several advantages over injection and oral administration, such as cost-effectiveness, low-stress induction, low labor costs, suitability for small fish, and the ability to vaccinate numerous simultaneously [17,43,44]. However, some crucial disadvantages remain, especially the need for high amounts of vaccines, weak immune responses, and low protection in small fish [45,46]. Additionally, the greatest limitation of this strategy is that the most effective age and size of fish for immersion vaccination are unclear in many fish species, including Nile tilapia.

Similar to other higher vertebrates, early immune responses in fish can be studied by examining the development of immune-related organs. Unfortunately, very few studies have shown early immune responses in Nile tilapia.

Previously, Cao et al. (2017) [47] investigated the development and differentiation mechanisms of the thymus gland using specific hybridization of the *RAG-1* gene in the early larval stage to the reproductive stage at 2 days post-fertilization (dpf) through 1 year of age. Importantly, it has been observed that the thymus gland of tilapia starts developing at an early stage, at approximately 2 and 5 dpf; the thymus gland grows according to the number of thymic cells, and at 7 and 10 dpf, the thymus gland shows a noticeable increase in size and a significant increase in the number of thymic cells. In addition, the thymus forms as a bulge on the surface dorsolateral to the pharyngeal cavity at those stages [48]. Furthermore, variations in the Nile tilapia embryo and larval development rates were observed, and a prominent thymus gland was present at 100 h. Hematopoietic tissue develops near the pronephros during early larval development [49]. In the case of Nile tilapia broodstock and larval immunization, Pasaribu et al. (2018) evaluated the efficacy of maternal transfer and offspring protection resulting from immunization with monovalent and bivalent vaccines (Biv). The relative percent survival of the Biv group revealed significantly enhanced disease resistance against *S. agalactiae* and *A. hydrophila* individually and under coinfection [50]. However, these results are too few to effectively apply to Nile tilapia vaccination. Therefore, our present study focused on assessing specific IgM antibody responses, analyzing the expression of specific immune genes under vaccination conditions, and evaluating vaccine efficacy through challenge tests.

In the current study, we detected the presence of IgM molecules via ELISA and immunohistochemistry and determined the expression patterns of immune-related genes, including *IgM*, *IgT*, *IgD*, *MHCIIα*, *TCRβ*, and *CD4*, which are good indicators of specific immune systems [51]. The results showed that larvae at 21 DAYC (0.108 ± 0.110 g) constitute the earliest stage for the effective expression of specific immune components post-immersion immunization with inactivated *S. agalactiae* vaccine, which suggests that this is an immunocompetent stage of Nile tilapia.

Fish in tropical areas generally reach the immunocompetent stage faster than fish in subtropical and temperate zones [52]. It takes approximately 28–30 days after hatching for fish to develop full immunocompetence. This is often marked by the appearance of the thymus, a primary lymphoid organ, and other immune-related cells and organs [53]. In Nile tilapia, the thymus is known to initially develop as early as two days after hatching, but it remains very small. As fish move into the immunocompetent stage, the thymus continues to grow progressively during the juvenile phase and then decreases in size as the animals age and their spines develop [47]. The results of this research and the findings from previous studies suggest that the age range of 28–30 days after hatching is a critical time when the immune system reaches a stage of full competence [53]. This stage signifies the transition from an immature to a fully functional immune system. However, the above information is insufficient to clearly describe the immunocompetent stage in Nile tilapia.

In this study, the fish at 28–42 DAYC (0.33–0.58 g) showed a remarkably rapid response compared to those at 21 DAYC (0.11 g). It is possible that the fish in this age group had more developed immune systems that enabled them to respond more effectively to the vaccines than those at 21 DAYC. Compared to the results of other studies in which fish were exposed to antigens via immersion vaccination, Nile tilapia of various ages or weights (sizes) may produce different specific antibodies. For example, in rainbow trout (*Oncorhynchus mykiss*), an increase in the levels of antibodies to *Vibrio anguillarum* was observed in juvenile fish weighing approximately 0.14 g or 2 weeks after hatching [54]. Young Nile tilapia weighing 0.1 ± 0.01 g could efficiently form antibodies against *S. agalactiae* [55]. A significant increase in antibody levels was observed in channel catfish (*Ictalurus punctatus*) after vaccination with *Edwardsiella ictalurii* compared with the control group. The first significant increase was observed in fish aged 4 weeks, with an average weight of 0.085 g [56]. In Asian seabass (*L. calcarifer*) at 35 and 42 dph with weights of approximately 0.10 ± 0.03 and 0.25 ± 0.14 g, respectively, a statistically significant increase in specific *IgM* levels against *S. iniae* was observed after administration of the heat-killed *S. iniae* vaccine after 7 days [57].

The expression of genes associated with the specific immune response of Nile tilapia larvae at 21, 28, 35, and 42 DAYC and at 24, 168, and 336 h after stimulation with FKV-SA was investigated. Normally, TCRβ is the receptor protein of T lymphocytes, while CD4 is essential in verifying the accuracy of binding between the TCRαβ-epitope and MHCIIαβ. Moreover, MHCIIαβ presents the structure of epitopes, which are partial components of pathogens or antigens, and their bond to T cells or B cells to stimulate these cells to produce cytokines. This stimulation involves T cells, B cells, or APCs, which trigger proliferation and differentiation processes to become memory cells, such as memory T or B cells, to remember pathogens or antigens of the same type or plasma B cells for the production of immunoglobulins (Ig) against specific diseases and antigens [58,59,60]. Based on this study, the expression of the *TCRβ*, *CD4*, and *MHCII*α genes in Nile tilapia fry at 21, 28, 35, and 42 DAYC that had received FKV-SA showed complete expression at all time intervals, especially after 336 hav. This expression was significantly higher than that in the unvaccinated control group. The immunoglobulin (Ig) types found in Nile tilapia with bony structures include IgM, IgT, and IgD [61]. The secreted immunoglobulin molecule (sIg) is critical in processes, such as neutralization, opsonization, antibody-dependent cytotoxicity, and complement activation. These components maximize the efficiency of disease and antigen control or the immune response [62]. We found that the expression of the *IgM* and *IgT* genes in Nile tilapia larvae immunized with FKV-SA at 21, 28, 35, and 42 DAYC was higher at all time intervals compared with that in unvaccinated fish, especially at 336 hav. In contrast, the *IgD* gene was specifically expressed in tilapia at 35 and 42 DAYC, with higher expression observed in the vaccinated group at 336 hav. The results suggest that IgM and IgT of Nile tilapia are crucial Igs in the response to invading pathogens and the protection of fish at early stages. Although IgD was found to be a late responder, the results suggest that it is a non-readiness molecule during the early stage of fish development. Similar to in higher vertebrates, fish IgD is a mysterious isotype that locates on the surface of immature B cells and is usually co-expressed with IgM to signal the B cells to be activated and ready to take part in the defense components of the immune system [61]. Therefore, the development of anti-IgT and anti-IgD antibodies for use in Nile tilapia is needed to further understand this phenomenon.

Additionally, the vaccine was effective in upregulating several immune-related genes, such as *TCRβ*, *CD4*, *MHCIIα*, *IgM*, *IgT* and IgD, in immune-related tissues, including the head kidney, spleen, PBLs, and gills, which is well supported by previous studies and other publications [63,64]. For example, in Asian seabass (*Lates calcarifer*), at 35 and 42 days posthatching (dph) and 7 days after vaccination, there was a statistically significant increase in the expression of the *CD4*, *MHCIIα*, *IgM*, *IgT*, and *IgD* genes [57]. In juvenile Nile tilapia, various nanovaccines were used to treat francisellosis and columnaris via immersion. It was observed that the expression of the *TCRβ*, *CD4*, *MHC IIα*, *IgM*, and *IgT* genes significantly increased in the head kidney, spleen, gills, and peripheral blood leukocytes (PBLs) 8 weeks after the first vaccine administration [19].

After immersion vaccination in tilapia of different ages, based on immunohistochemistry at 21, 28, and 35 DAYC, IgM was found to be distributed in different organs. The most densely distributed IgM was found in the head kidney and intestine. The organ with the highest IgM reaction was the head kidney, which is consistent with the results observed in *Takifugu rubripes* in all organs, with higher expression observed in the primary lymphoid organs, such as the head kidney and spleen [65]. In *Pseudosciaena crocea* and *Larimichthys crocea*, significant *IgM* heavy chain gene expression was detected in the spleen, peripheral blood, and head kidney [66,67]. Furthermore, using the immunohistochemistry technique in Nile tilapia after receiving a killed *S. agalactiae* vaccine, the distribution of IgM was observed across various organs of the fish, with a prominent presence in the head kidney [55].

The above-obtained information strongly supports the phenomena found in the challenge test experiment. When the 21–42 DAYC larvae were immunized with FKV-SA, the 21 DAYC larvae showed significantly higher differences in survival than the control at only 336 hav, which was different from the results obtained for 28–42 DAYC larvae. In these groups, fish larvae immunized with FKV-SA showed significantly higher survival than unvaccinated fish in the presence of viable *S. agalactiae* at 168 and 336 hav, which was 7 days earlier than what was observed in 21 DAYC larvae. This indicated that the optimal period for initial vaccination in Nile tilapia larvae is 28–35 DAYC.

This study revealed that administering FKV-SA to Nile tilapia larvae from 21 DAYC onward can effectively stimulate a specific and targeted immune response against *S. agalactiae* infection. This success can be attributed to the role of both primary and secondary lymphoid organs and tissues, particularly the various types of mucosa-associated lymphoid tissues (MALTs). These tissues are characterized by their mucous structure and their involvement in the general immune response. Fish possess mucous tissues that cover every surface of their bodies, and thus, the structure of these mucous tissues surrounding the fish body is crucial in the formation of a specific immune system capable of detecting environmental disturbances. They serve as the first line of defense against disease pathogens, enabling fish to interact with the external environment that surrounds them. In particular, gill-associated lymphoid tissue (GIALT) is essential. Additionally, lymphoid tissue in the skin, referred to as skin-associated lymphoid tissue (SALT), has also been reported to harbor components related to immunity. These findings have confirmed that these tissues can indeed trigger responses against diseases and antigens [68,69,70,71].

Although the vaccination method used in this study involved immersion, there are reports indicating that immersion vaccination can stimulate immune system activity in the GALT within the gut. It has been reported that B cells, particularly IgM and IgT B cells, are found in the lamina propria (LP) layer of the intestine, which is a part of the gut mucosa [72,73]. Additionally, because freshwater fish do not drink water to regulate their body’s salt and mineral balance (osmoregulation) but rather engage in sipping behaviors, there is a chance that the vaccine, dissolved in water, can enter the digestive system. This can lead to stimulation of the GALT in the gastrointestinal tract of the fish, similar to the process with immersion vaccines. The fish digestive system is considered a crucial factor that is directly interconnected with the external environment through ingestion. This process provides a pathway for potential disease pathogens and antigens to enter the fish body. Moreover, it serves as a pathway for triggering specific immune responses, especially in the mucosal immune system, such as in the GALT within the intestine. The GALT is crucial in maintaining the gut equilibrium and acts as a defense mechanism against various diseases and foreign agents [74].

In the study of disease resistance against *S. agalactiae* in Nile tilapia larvae at 21, 28, 35, and 42 DAYC and at 24, 168, and 336 h post-vaccination with the FKV-SA vaccine, the post-vaccination survival rates between the vaccinated and control groups showed significant differences starting at 21 DAYC at 336 h after vaccination. However, for 28, 35, and 42 DAYC, the vaccinated fish exhibited significant disease resistance compared to the control group at 168 and 336 h after vaccination across all age groups. Furthermore, the RPS values were found to differ significantly between vaccinated and nonvaccinated fish in each age group (21–42 days). However, the RPS values obtained ranged from relatively low values of 28.14 to 45.54. This suggested that while the fish in these age ranges had developed specific immune responses to protect against and resist *S. agalactiae* infection, the effectiveness of the immune response was not very high. Generally, a level of 60 or higher is considered acceptable [75].

Based on this, more suitable vaccine formulations should be developed. Studies have suggested that the effectiveness of immune responses can be improved through vaccination methods, such as immersion, which enhances the overall immune response. For instance, the utilization of mucoadhesive cationic lipid-based nanoencapsulation for vaccines to prevent columnaris in Asian seabass (*Lates calcarifer*) through immersion resulted in a survival rate of 72.50 ± 3.54% and the highest RPS recorded (62.07 ± 4.87). Similarly, the application of nanovaccines to combat francisellosis and columnaris through immersion in juvenile Nile tilapia has exhibited survival rates ranging from 65.83 to 72.50% and RPS values ranging from 52.87 to 62.07 [3,19]. Furthermore, the utilization of mucoadhesive polymers to enhance the effectiveness of inactivated vaccines in preventing columnaris caused by *F. columnare* infection in red tilapia has shown a maximum survival rate and RPS of 83% and 81, respectively [76]. These results showcase immersion as an additional approach to enhance the efficacy of inactivated vaccines for preventing harmful bacterial diseases.

Finally, the results from the current study demonstrated that specific immune responses start developing in Nile tilapia larvae from 21 DAYC onwards. The knowledge gained from this study is crucial and can be applied to the development of vaccines in small Nile tilapia, enabling the effective prevention of streptococcosis, a disease commonly occurring in the fish-farming industry.

## 5. Conclusions

Administration of the *S. agalactiae* vaccine (FKV-SA) to Nile tilapia larvae via immersion initially induced the expression of a specific IgM and of targeted immune-related genes in the fish at as early as 21 DAYC. It is possible that the fish completes its developmental cycle and reaches maturity at approximately 21 DAYC, indicating the critical period when Nile tilapia larvae develop immunocompetence. Based on the expression data for specific key genes and challenge tests against viable *S. agalactiae*, we believe that the optimal vaccination period for Nile tilapia larvae is between 28 and 35 DAYC. However, the RPS of vaccinated larvae at various DAYCs was relatively low. This result suggests that although vaccination can elicit a crucial immune response in fish, protection against disease is relatively inefficient. This finding could have been obtained due to several factors, such as low immunogenicity of the FKV-SA form, low antigen uptake via immersion administration, or immature cell-mediated immunity (not investigated in this study), and further research and development is needed in the future. Nonetheless, the results of this study provide important foundational knowledge that can be used to develop effective vaccination methods to reduce constraints and losses in Nile tilapia production caused by severe and highly damaging disease outbreaks. Early vaccination may contribute to the sustainable, nonantibiotic farming of Nile tilapia soon.

## Figures and Tables

**Figure 6 vaccines-11-01753-f006:**
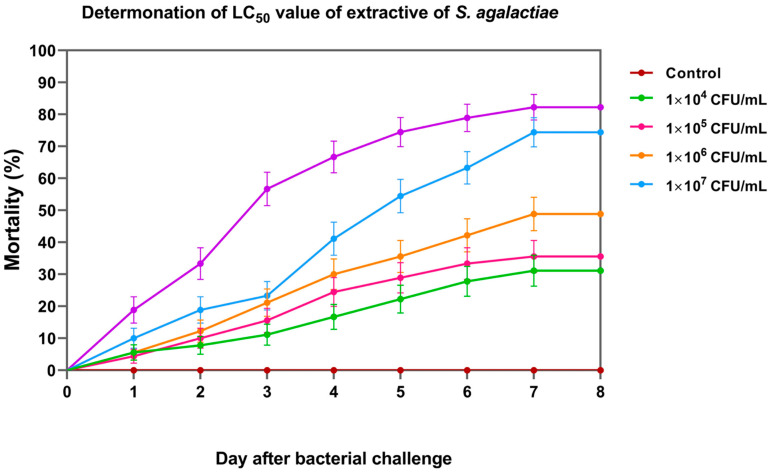
Cumulative mortality of Nile tilapia larvae exposed to different concentrations of *S. agalactiae* for median lethal concentration (LC_50_) analysis.

**Figure 7 vaccines-11-01753-f007:**
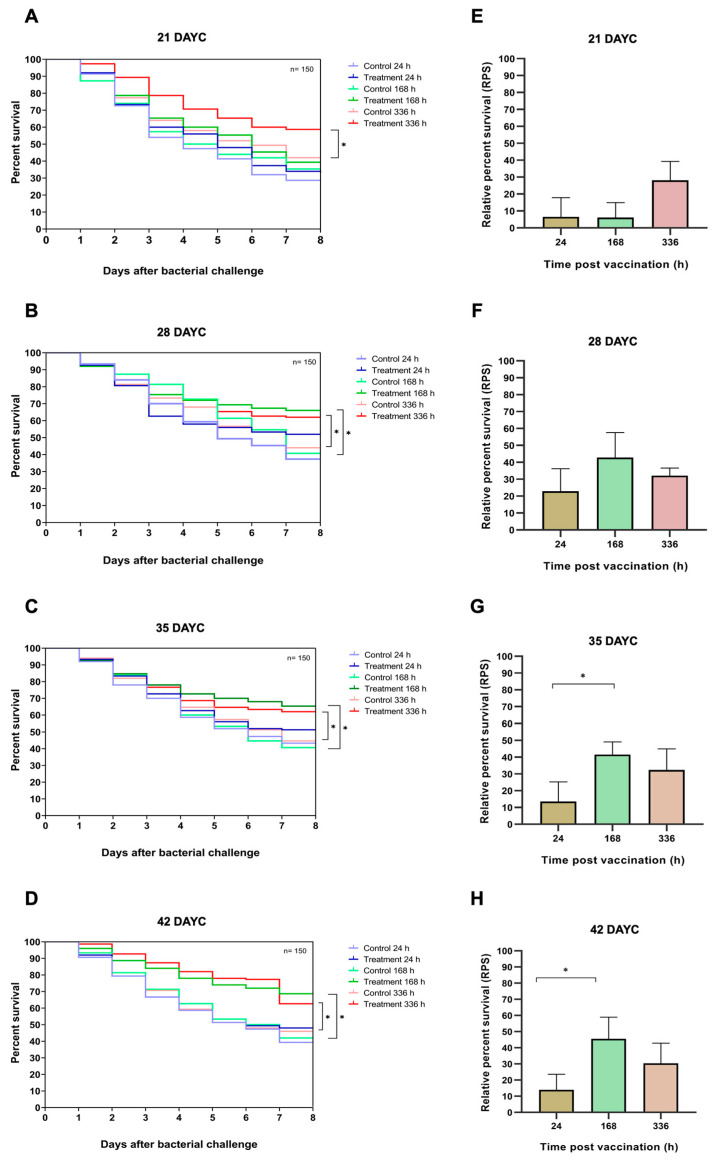
Data and survival plots were generated using the Kaplan–Meier method (**A**–**D**). Survival analysis and relative percent survival (RPS) of Nile tilapia FKV-SA vaccinated after challenges with *S. agalactiae* at 21, 28, 35, and 42 DAYC (**E**–**H**). The levels of statistical significance between the control and treatment groups are indicated by * (*p* < 0.05).

**Table 1 vaccines-11-01753-t001:** Primers used in this study to determine gene expression levels in Nile tilapia (*Oreochromis niloticus*).

Gene	Primer Name	Nucleotide Sequences (5′ → 3′)	Amplicon Size (bp)	Tm (°C)	References
β-Actin	On_β-actin	F: ACAGGATGCAGAAGGAGATCACAG R: GTACTCCTGCTTGCTGATCCACAT	155	60	[19]
T-cell receptor β-chain (constant region)	On_TCRβ	F: GGACCTTCAGAACATGAGTGCAG R: TCTTCACGCGCAGCTTCATCTGT	164	60	[22]
Cluster of differentiation 4 (*CD4*)	On_CD4	F: GCTCCAGTGTGACGTGAAA R: TACAGGTTTGAGTTGAGCTG	150	60	[19]
Major histocompatibility complex (*MHC*) class IIα molecules	On_MHC-IIα	F: CAGTGTTTGATGTGTTTTCAG R: CTCTTCACCATCCAGTCCA	100	60	[19]
Immunoglobulin M heavy chain (*IgHM*)	On_IgM	F: GGATGACGAGGAAGCAGACT R: CATCATCCCTTTGCCACTGG	122	60	[19]
Immunoglobulin T heavy chain (*IgHT*)	On_IgT	F: TGACCAGAAATGGCGAAGTCTG R: GTTATAGTCACATTCTTTAGAATTACC	136	60	[19]
Immunoglobulin D heavy chain (*IgHD*)	On_IgD	F: AACACCACCCTGTCCCTGAAT R: GGGTGAAAACCACATTCCAAC	127	60	[23]

## Data Availability

The data that support the findings of this study are available on request from the corresponding authors.

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
