# Peer review of "First Investigation of the Optimal Timing of Vaccination of Nile Tilapia (Oreochromis niloticus) Larvae against Streptococcus agalactiae"

_vaccines, 2023, doi:10.3390/vaccines11121753_

Round 1

Reviewer 1 Report

Comments and Suggestions for Authors

This manuscript described the optimal vaccination timing for tilapia  larvae against Streptococcus agalactiae. The work is interesting  and provide a reference for the vaccination of fish. But some issues are needed to address.

1. The study just investigated the IgM in protein level, how about the other Ig?

2. In general, the highest Ig M level appares at 28 days post vaccination, but the result in this study is different, why?

3.  The expression level of CD8 should be examined;

4. The expression levels of cytokines involved in innate immunity should be examined;;

5. The images of histological analysis are not clear enough. 

Comments on the Quality of English Language

Moderate editing of English language required. 

Author Response

Dear esteemed editor and reviewers

Firstly, we appreciate all the comments suggested in our manuscripts. These comments are beneficial for improving the quality of our scientific research to warrant an acceptable research document.

In the current version of our manuscript, we carefully correct and add necessary information by strictly following the valuable suggestions of the 3 anonymous reviewers, indicated by green highlight as follows;    

Reviewer 1

This manuscript described the optimal vaccination timing for tilapia larvae against Streptococcus agalactiae. The work is interesting and provides a reference for the vaccination of fish. But some issues are needed to address.

  1. The study just investigated the IgM in protein level; how about the other Igs?

Response: Thank you so much for this comment. We understand this excellent comment and the good intentions of the reviewer. Since we have some limitations of other Igs detection, we conducted ELISA for IgM measurements based on our monoclonal antibody resources. However, with this limitation, we designed to employ qRT-PCR analysis to compensate for and indicate expression levels of all Igs at transcriptional levels. By the way, we will take this excellent comment to improve our further study.      

  1. In general, the highest Ig M level appears at 28 days post-vaccination, but the result in this study is different; why?

Response: Thank you so much for this comment. In our experiment, we did not get fish exposed to the SA-killed vaccine for an extended period of 28 days. At the same time, we were targeting to demonstrate IgM expression in each DAYC period. In the details, our results indicated that at 28 DAYC, the expression level of IgM was higher than that of 21 DAYC but lower than those found at 35 DAYC and 42 DAYC after exposure with SA killed vaccine 168-336 hours.

  1. The expression level of CD8 should be examined.

Response: Thank you so much for this comment; we agreed well with this comment. We thought CD4+ T cells should be more dominant than CD8+ T cells to drive immune responses against bacterial infection. However, we will take this excellent comment for our further study.      

  1. The expression levels of cytokines involved in innate immunity should be examined.

Response: Thank you so much for this comment. We also agreed well with this comment. Our current research emphasizes cytokine responses during vaccination in larval stages of Nile tilapia, and we hope to launch this information via publication very soon. 

  1. The images of histological analysis are not clear enough. 

Response: Thank you so much for this comment; we also agreed with this comment. We tried to replace it with better histological alteration information in our manuscript's current version.

Reviewer 2 Report

Comments and Suggestions for Authors The present manuscript by Kumwan et al., titled “First investigation of the optimal vaccination timing for Nile tilapia (Oreochromis niloticus) larvae against Streptococcus agalactiae” investigated early immune responses and explored the optimal vaccination periods in Nile tilapia post S. agalactiae infection in lab conditions. The results and findings are useful to researchers working on fish vaccines. The author has used immune response techniques to address the research questions. The MS requires comments which need to be addressed in the present version as mentioned below;

It is advisable to discuss SA pathogenesis in fish in the introduction section.

It is important to highlight why did author considered immersion vaccine to the larval stages of tilapia.

section 2.3.3- the author must change serum to total IgM because the author measured antibody from total supernatant instead of serum. 

section 2.5 Statistical analysis. Did the author conduct homogeneity of variance before ANOVA?

section 2.3.6 For the gene expression authors used a single HKG for normalisation. It is recommended to use more than 2 HKG for validation.

what is the source of IgM for immunohistochemical analysis?

The conclusion must be mentioned according to the results obtained based on treatment and immune response instead of generalised Comments on the Quality of English Language

Minor editing required

Author Response

Dear esteemed editor and reviewers

Firstly, we appreciate all the comments suggested in our manuscripts. These comments are beneficial for improving the quality of our scientific research to warrant an acceptable research document.

In the current version of our manuscript, we carefully correct and add necessary information by strictly following the valuable suggestions of the 3 anonymous reviewers, indicated by green highlight as follows;    

Reviewer 2

The present manuscript by Kumwan et al., titled “First investigation of the optimal vaccination timing for Nile tilapia (Oreochromis niloticus) larvae against Streptococcus agalactiae” investigated early immune responses and explored the optimal vaccination periods in Nile tilapia post-S. agalactiae infection in lab conditions. The results and findings are useful to researchers working on fish vaccines. The author has used immune response techniques to address the research questions. The MS requires comments which need to be addressed in the present version as mentioned below;

It is advisable to discuss SA pathogenesis in fish in the introduction section.

Response: Thank you so much for this critical point; we have added this information in the “Introduction”.

It is important to highlight why did author considered immersion vaccine to the larval stages of tilapia.

Response: Thank you so much for this critical point; we have added this information in both “Introduction” and “Discussion”.

Section 2.3.3- the author must change serum to total IgM because the author measured antibody from total supernatant instead of serum.

It is important to highlight why did author considered immersion vaccine to the larval stages of tilapia.

Response: Thank you so much for this point; we have changed this information in this section.

Section 2.5 Statistical analysis. Did the author conduct homogeneity of variance before ANOVA?

Response: Thank you for this critical point; we have indicated this information in this section.

Section 2.3.6 For the gene expression authors used a single HKG for normalisation. It is recommended to use more than 2 HKG for validation.

Response: Thank you so much for this critical point. We understand well intention of the reviewer well. In our lab, we have tested the precision of qRT-PCR using various housekeeping genes, and we have experienced that beta-actin is the most stable in various stages and conditions in Nile tilapia experiments.   

What is the source of IgM for immunohistochemical analysis?

Response: Thank you so much for this critical point. We clarified this unclear point in this section.

The conclusion must be mentioned according to the results obtained based on treatment and immune response instead of generalised.

Response: Thank you so much for this point.  We have carefully modified this part to meet the reviewer's suggestions.

Reviewer 3 Report

Comments and Suggestions for Authors

Manuscript ID: vaccines-2657163

Title: First investigation of the optimal vaccination timing for Nile tilapia (Oreochromis niloticus) larvae against Streptococcus agalactiae

Authors: Benchawan Kumwan , Anurak Bunnoy , Satid Chatchaiphan , Pattanapon Kayansamruaj , Ha Thanh Dong , Saengchan Senapin * , Prapansak Srisapoome

The manuscript investigates the early immune responses of Nile tilapia after immersion vaccination with inactivated S. agalactiae to develop an optimal vaccination strategy in this species. The investigation of an early immune response included the assessment of IgM in tissues, expression of immune-relate genes and challenge tests.    

The study is adding new information and is of interest to the readership of Vaccines.

However, there are several major and minor issues to address.

General issues

Generally, there are many badly formulated sentences that make the manuscript difficult to read. Therefore, as a primary thing, I suggest the authors to considerably improve the language throughout the manuscript.

This is a study with quite complicated design and the authors are not explaining the experimental setup clearly. Therefor, a figure illustrating the study design would be helpful.     

Specific issues

Line 91: How was the pathogen testing done?

Line 131: Delete the redundant 12 min

131: The authors mention that “Bacteria cell density was measured by previous methods”. Can the authors specify those methods here? Considering that previously mentioned methods are OD measurement and CFU count, how is the CFU count done on formalin-inactivated bacteria?

Again, the experimental design section 2.3.1. is not described clearly. This section would benefit from a schematic overview of the tanks, fish number and different groups. In addition, the schematic overview of sampling could be added and the text should include references to the tables and figures.      

For example, in line 141, the authors write that 7 different Nile tilapia larvae were used at 1, 7, 21, 28, 35, 42 DAYC. Is this correct or were there 7 different groups of Nile tilapia used in this study?

Line 143: “Nile tilapia larvae were randomly divided from storage tanks….”. How many storage tanks were there? In Line 99, the authors mention only one 3000L tank with 7000 larvae.

Line 145: How many fish were in the 28 experimental tanks?

Line 163: Can the authors describe the serum collection for ELISA?

It is unclear what did the authors test with ELISA, was it serum or the supernatant from the whole larvae (as mentioned in line 185). I suggest the authors to specify that. Also, was each fish tested individually or did the authors pool the groups?      

Line 182: Correct “cells” to “wells”

Line 203: What visceral organs were sampled? In line 161, the authors mention that the whole fish were kept in TRIzol. In line 205, the authors mention that only the tissues (visceral organs) were sampled. Can the authors specify this?

Line 206: Please provide more information regarding the RNA purification and the kit used.

2.3.7. Immunohistochemical analysis.

How were the differences in IgM signal evaluated? Were the positive cells counted? if yes, then in which area? The authors need to provide a thorough description of how were the differences in IgM cells/signal between different groups assessed, i.e. method of quantification.    

Line 247: Please correct that the samples were deparaffinized in xylene and rehydrated in graded ethanol series.

The "Challenge test of vaccinated Nile tilapia larvae" section 2.4.2. is not clear and should be re-written for clarity. For example, the title states that the larvae were vaccinated. However, this study also included unvaccinated control fish.     

Line 282: Please specify how many larvae per experimental group

Line 283: Please specify that the age is DAYC

Line 298: Add tank after 250 L glass

Line 333: The authors mention that the gene expression was analyzed in the whole body of larvae, but in line 203, it is stated the visceral organs were sampled for gene expression. Which is correct?   

Figure 5. The letters in the figures are not always visible against the dark background. Also, please add a legend for yellow arrows.

The section 3.4. “Weight and length measurement” is redundant. This data is already presented in table S1 and S2.

Line 484: "Unacceptable" might not be the correct adjective to use here. Injection-vaccines are still in use for many fish species and often the only route of infection offering a sufficient degree of protection.

The discussion part also needs to be corrected for language deficits.

Line 582: Please specify here that it’s the anti-IgD/IgT antibodies that the authors are referring to.

Line 611: Please delete “after vaccination”.

Line 634: Do the authors mean GALT within the gut?

Comments on the Quality of English Language

The manuscript should be corrected for language.

Author Response

Dear esteemed editor and reviewers

Firstly, we appreciate all the comments suggested in our manuscripts. These comments are beneficial for improving the quality of our scientific research to warrant an acceptable research document.

In the current version of our manuscript, we carefully correct and add necessary information by strictly following the valuable suggestions of the 3 anonymous reviewers, indicated by green highlight as follows;    

Reviewer 3

The manuscript investigates the early immune responses of Nile tilapia after immersion vaccination with inactivated S. agalactiae to develop an optimal vaccination strategy for this species. The investigation of an early immune response included the assessment of IgM in tissues, expression of immune-related genes and challenge tests.   

The study is adding new information and is of interest to the readership of Vaccines.

However, there are several major and minor issues to address.

General issues

Generally, there are many badly formulated sentences that make the manuscript difficult to read. Therefore, as a primary thing, I suggest the authors to considerably improve the language throughout the manuscript.

Response: Thank you so much for this comment. Our manuscript has been intensively edited in this version by a scientist keen on this field.

This is a study with quite complicated design and the authors are not explaining the experimental setup clearly. Therefore, a figure illustrating the study design would be helpful.    

Response: Thank you so much for this comment. In this version, our manuscript has properly illustrated all experimental designs in the manuscript via the graphical abstract.

Specific issues

Line 91: How was the pathogen testing done?

Response: Thank you so much for this comment. We indicated microbiological techniques used for pathogen detection in the used broodstocks.

Line 131: Delete the redundant 12 min.

Response: Thank you so much for this comment. We deleted this redundant component.

131: The authors mention that “Bacteria cell density was measured by previous methods”. Can the authors specify those methods here? Considering that previously mentioned methods are OD measurement and CFU count, how is the CFU count done on formalin-inactivated bacteria?

Response: Thank you so much for this comment. We simplified this sentence in the manuscript. We used absorbance 1.0 at OD600 to prepare bacterial cell concentration at 1 × 109 CFU/mL   

Again, the experimental design section 2.3.1. is not described clearly. This section would benefit from a schematic overview of the tanks, fish number and different groups. In addition, the schematic overview of sampling could be added and the text should include references to the tables and figures. 

For example, in line 141, the authors write that 7 different Nile tilapia larvae were used at 1, 7, 21, 28, 35, 42 DAYC. Is this correct or were there 7 different groups of Nile tilapia used in this study?

Response: Thank you so much for this comment. This information needs to be clearly illustrated as described above.  

Line 143: “Nile tilapia larvae were randomly divided from storage tanks….”. How many storage tanks were there? In Line 99, the authors mention only one 3000L tank with 7000 larvae.

Response: Thank you so much for this comment. This information needs to be clearly illustrated as described above.  

Line 145: How many fish were in the 28 experimental tanks?

Response: Thank you so much for this comment. This information needs to be clearly illustrated as described above.  

Line 163: Can the authors describe the serum collection for ELISA?

Response: Thank you so much for this comment. In this part, we should have corrected the fish serum. The whole fish larvae extracted total soluble proteins with the methods described in [19]. This information needs to be adequately clarified.

It is unclear what did the authors test with ELISA, was it serum or the supernatant from the whole larvae (as mentioned in line 185)? I suggest the authors to specify that. Also, was each fish tested individually or did the authors pool the groups?

Response: Thank you so much for this comment. This information needs to be clarified adequately in this part.

Line 182: Correct “cells” to “wells”.

Response: Thank you so much for this comment. We corrected this error point.

Line 203: What visceral organs were sampled? In line 161, the authors mention that the whole fish were kept in TRIzol. In line 205, the authors mention that only the tissues (visceral organs) were sampled. Can the authors specify this?

Response: Thank you so much for this comment. We clarified this unclear point in the manuscript.

Line 206: Please provide more information regarding the RNA purification and the kit used.

Response: Thank you so much for this comment. We clarified this unclear point in the manuscript.

2.3.7. Immunohistochemical analysis.

How were the differences in IgM signal evaluated? Were the positive cells counted? if yes, then in which area? The authors need to provide a thorough description of how were the differences in IgM cells/signal between different groups assessed, i.e. method of quantification.   

Response: Thank you so much for this comment. We clarified this unclear point in the manuscript.

Line 247: Please correct that the samples were deparaffinized in xylene and rehydrated in graded ethanol series.

Response: Thank you so much for this comment. We corrected this error information.

The "Challenge test of vaccinated Nile tilapia larvae" section 2.4.2. is not clear and should be re-written for clarity. For example, the title states that the larvae were vaccinated. However, this study also included unvaccinated control fish.

Response: Thank you so much for this comment. We corrected this error information.   

Line 282: Please specify how many larvae per experimental group.

Response: Thank you so much for this comment. We corrected this error information.   

Line 283: Please specify that the age is DAYC.

Response: Thank you so much for this comment. We corrected this error information.

Line 298: Add tank after 250 L glass

Response: Thank you so much for this comment. We corrected this error information.

Line 329: The authors mention that the gene expression was analyzed in the whole body of larvae, but in line 203, it is stated the visceral organs were sampled for gene expression. Which is correct?

Response: Thank you so much for this comment. Our experiment used the whole body for ELISA and immune-gene expression analysis. Meanwhile, visceral organs were preserved for immunohistochemistry investigation. We corrected this error information.

Figure 5. The letters in the figures are not always visible against the dark background. Also, please add a legend for yellow arrows.

Response: Thank you so much for this comment. We corrected this error information.

The section 3.4. “Weight and length measurement” is redundant. This data is already presented in table S1 and S2.

Response: Thank you so much for this comment. We appropriately modified this information in this part.

Line 491: "Unacceptable" might not be the correct adjective to use here. Injection-vaccines are still in use for many fish species and often the only route of infection offering a sufficient degree of protection.

Response: Thank you so much for this comment. We slightly modified it to create a better flow.

The discussion part also needs to be corrected for language deficits.

Response: Thank you so much for this comment. We carefully revised this part based on the suggestions of the native speaker, who helped us improve the language quality.

Line 589: Please specify here that it’s the anti-IgD/IgT antibodies that the authors are referring to.

Response: Thank you so much for this comment. We carefully revised this part based on the reviewer's suggestions.

Line 611: Please delete “after vaccination”.

Response: Thank you so much for this comment. We deleted this content from the manuscript.

Line 634: Do the authors mean GALT within the gut?

Response: Thank you so much for this comment. We corrected this error.

Round 2

Reviewer 1 Report

Comments and Suggestions for Authors

The revised manuscript is much better and should be considered for publication.

Author Response

Dear esteemed editor and reviewers

Firstly, we appreciate all the comments suggested in our manuscripts. These comments are beneficial for improving the quality of our scientific research to warrant an acceptable research document.

In the current version of our manuscript, we carefully correct and add necessary information by strictly following the valuable suggestions of the reviewers, indicated by blue highlight as follows;    

Reviewer 1

This paper makes an important, albeit incremental advance in our understanding of fish immunity and vaccine responses.

I have not found adequate responses to the points copied below made by reviewer 3, including the provision of a schematic for a part of the study.

However, Reviewer 3 is now unavailable although the authors have responded to some of their comments, I have not found adequate responses to the points copied below, including the provision of a schematic for a part of the study.

Again, the experimental design section 2.3.1. is not described clearly. This section would benefit from a schematic overview of the tanks, fish number and different groups. In addition, the schematic overview of sampling could be added and the text should include references to the tables and figures.

For example, in line 141, the authors write that 7 different Nile tilapia larvae were used at 1, 7, 21, 28, 35, 42 DAYC. Is this correct or were there 7 different groups of Nile tilapia used in this study?

Line 143: “Nile tilapia larvae were randomly divided from storage tanks….”. How many storage tanks were there? In Line 99, the authors mention only one 3000L tank with 7000 larvae.

Line 145: How many fish were in the 28 experimental tanks?

Response: Thank you so much for these critical comments. We have carefully corrected all information important for describing these concerns in section 2.31. Additionally, figure legends for supplement material Figure 1 and 2 have been provided in this version to increase better flow of the manuscript.

Line 163: Can the authors describe the serum collection for ELISA?

Response: Thank you so much for this comment. In this part, we should have corrected the fish serum. The whole fish larvae extracted total soluble proteins with the methods described in [19]. This information needs to be adequately clarified.

It is unclear what did the authors test with ELISA, was it serum or the supernatant from the whole larvae (as mentioned in line 185)? I suggest the authors to specify that. Also, was each fish tested individually or did the authors pool the groups?

Response: Thank you so much for these concerns. In section 2.3.5, we have clarified the specific protocol used to extract the supernatant, which is further used for ELISA assay.

I also have the following additional comments:

In Figure 1-4, the way that the statistical significance is ascribed on the figures definitely needs to be adjusted. The use of a lettering system is unnecessarily complex and not easily understood! suggest showing the actual p values only for differences which are significant with bars indicating the comparisons made.

Response: Thank you so much for these concerns. We well understand the concerns regrading on this point. However, in the basic analysis of factorial designs, combinatorial factors were conducted based on 2×5 combinatorial factors generated by 2 vaccination treatments and 5 different time courses. Therefore, it is important to demonstrate significant difference among all combinatorial factors to fit the criteria of the experimental designs.   

Further comment should be provided on the immunological and biological significance of the observed increases in IgD mRNA levels post vaccination and the potential role in immune regulation versus protection.

Response: Thank you so much for this comment. We have discussed this concerned point in discussion section.
